# The Therapeutic Effects of Oral Intake of Hydrogen Rich Water on Cutaneous Wound Healing in Dogs

**DOI:** 10.3390/vetsci8110264

**Published:** 2021-11-04

**Authors:** Dong-Dong Qi, Meng-Yuan Ding, Ting Wang, Muhammad Abid Hayat, Tao Liu, Jian-Tao Zhang

**Affiliations:** 1College of Veterinary Medicine, Northeast Agricultural University, Harbin 150030, China; qidongok@163.com (D.-D.Q.); 15776560839@163.com (M.-Y.D.); w1372368457t@163.com (T.W.); abidbagra.uvas@gmail.com (M.A.H.); liutaotiger@163.com (T.L.); 2Heilongjiang Key Laboratory for Laboratory Animals and Comparative Medicine, Harbin 150030, China

**Keywords:** antioxidant, dogs, hydrogen-rich water, wound healing

## Abstract

This study explored the effects of drinking Hydrogen-rich water (HRW) on skin wound healing in dogs. Eight circular wounds were analyzed in each dog. The experimental group was treated with HRW thrice daily, while the control group was provided with distilled water (DW). The wound tissues of dogs were examined histopathologically. The fibroblasts, inflammatory cell infiltration, the average number of new blood vessels, and the level of malondialdehyde (MDA) and superoxide dismutase (SOD) activity in the skin homogenate of the wound was measured using the corresponding kits. The expressions of Nrf-2, HO-1, NQO-1, VEGF, and PDGF were measured using the real-time fluorescence quantitative method. We observed that HRW wounds showed an increased rate of wound healing, and a faster average healing time compared with DW. Histopathology showed that in the HRW group, the average thickness of the epidermis was significantly lower than the DW group. The average number of blood vessels in the HRW group was higher than the DW group. The MDA levels were higher in the DW group than in the HRW group, but the SOD levels were higher in the HRW group than in the DW group. The results of qRT-PCR showed that the expression of each gene was significantly different between the two groups. HRW treatment promoted skin wound healing in dogs, accelerated wound epithelization, reduced inflammatory reaction, stimulated the expression of cytokines related to wound healing, and shortened wound healing time.

## 1. Introduction

As the largest organ in animals, the skin plays a critical role in preventing animals from pathogenic microorganisms and wounds [1]. Skin wounds caused by diverse elements are routinely presented in a clinical veterinary setting. The recovery of severe wounds associated with numerous factors, such as therapy duration and treatment strategies, might lead to various therapeutic effects affecting the time and effects of wound healing [2]. Wound healing involves a series of mechanisms, such as inflammatory response and oxidative stress, which play vital roles in wound pathophysiology [3]. Numerous therapeutic approaches have been used in clinical treatments, including photodynamic therapy, far-infrared ray therapy, adipose tissue extraction therapy, and arginine-based materials therapy [4,5,6,7], along with various biomaterial platforms based on anti-inflammation therapy strategies [8].

Previous studies have reported that hydrogen gas can exert therapeutic effects through anti-oxidation. Hydrogen gas has the lowest molecular weight, along with a strong antioxidant capacity to remove excessive reactive oxygen species (oxygen-free radicals) from the body [9]. In 1975, Dole M found that high-pressure hydrogen gas could effectively treat malignant skin tumors [10]. Clinical studies have shown that hydrogen therapy has therapeutic potential for the treatment of rheumatoid arthritis, diabetes, Parkinson’s disease, and metabolic syndrome [11,12]. Tomohiro et al. found that hydrogen molecules could activate the NF-E2-related factor (Nrf-2) antioxidant defense pathway and ameliorate hyperoxic lung injury via the induction of Nrf-2 dependent genes, such as HO-1 [13]. Several studies have suggested that the inhalation of hydrogen-containing gas constitutes a novel guideline for practical clinical care, focusing on the treatment of skin wounds [14].

Hydrogen gas is a flammable and explosive gas, which is dangerous and difficult to store; however, hydrogen gas-saturated water, which is known as hydrogen-rich water (HRW), is safe and easy to absorb, and is frequently used in medicine [13]. HRW has been reported to exhibit antioxidant and anti-inflammatory effects, reducing the levels of pro-inflammatory cytokines, effectively protecting against organ damage [15]. Hydroxyl radical and peroxynitrite (ONOO-) ion have strong oxidation toxicity, and the potential to cause lipid peroxidation, DNA damage, and protein peroxidation, resulting in cell damage [16]. However, hydrogen dissolved in water can react with them, reducing the consequent damage. When H_2_ enters the body and reacts, only water is produced, discharging the excess H_2_ from the body, implying that HRW could be safely used to treat diseases without any side-effects.

Previous studies on animal testing have used both oral ingestion of HRW and injection of hydrogen-rich saline (HRS) as major routes of administration of HRW [17]. Studies have shown that HRS can scavenge hydrogen peroxide and reactive oxygen species (ROS) [18,19] by upregulating the expression of heme oxygenase (HO-1) as well as the reduction of quinine (NQO-1) [20]. Additionally, recent studies have demonstrated the antifatigue effects of HRW in animals and humans [21]. HRW also induces anti-apoptotic effects in myocardial cells [22] and inhibits the production of α, β-dicarbonyl compounds and ROS in the kidneys of rats [23]. Another study suggested that the use of HRW could enhance the rate of palatal wound recovery [24]. Thus, HRW has been shown to be one of the underlying therapeutic strategies regarding wound treatments. Although HRW has been proven to accelerate the recovery of skin wounds [14], promote the healing of skin injury induced by radiotherapy in rats [25], there is little information available on the use of HRW in trauma associated with canine skin.

Here we aimed to explore whether HRW could play an essential role in the recovery of skin wounds in a canine model. This is the first veterinary research-based study to purposefully assess the effects of HRW on cutaneous wound healing in dogs. The specific objectives were (1) to compare the effect of the healing of open wounds in dogs treated with HRW and the ones treated with distilled water (DW); (2) to compare the level of oxidative stress and inflammatory indicators; (3) to explore whether or not HRW had therapeutic value in the recovery of canine skin wound. We hypothesized that HRW promoted angiogenesis and elevated tissue antioxidant gene expression to promote wound healing and accelerate the rate of wound healing, making it a safe and effective clinical treatment.

## 2. Materials and Methods

### 2.1. Dogs

Twelve healthy male beagle dogs (age: 2–4 years; weight: 9.0–11.3 kg) from a breeder based in Harbin and housed in a standard environment and provided free access to standard dog food and water. Food and water were withdrawn 6 h prior to anesthesia. All methods complied with good animal practices required by the Animal Ethics Procedures and Guidelines of the People’s Republic of China. We made every effort to minimize animal suffering and reduce the number of animals used. No deaths were reported in this study; both the mental status and diet of the dogs were normal. None of the animals were killed and all dogs were healthy and adopted by animal protection organization after finishing the experiment.

### 2.2. Creation of Surgical Wounds

Dogs were randomly divided into control and experimental groups (*n* = 6). The hair was clipped on the dorsolateral trunk on day 0. Cephalic intravenous catheter was placed, and the dogs were anesthetized using propofol (0.65 mg/kg), followed by the inhalation of 3.5% isoflurane via endotracheal intubation, and the physiological state of the experimental dogs was monitored in real time during the operation.

Four circular wounds with a diameter of 15 mm were created on each side of the midline of the shaved dorsum using a dermal biopsy punch. Each wound was 20 mm from the midline of the dorsum, and the adjacent wounds were 25 mm from each other. From the tail side to the head side, and from left to right, the wounds were labeled as A1, A2, A3, A4, B1, B2, B3, B4 (Figure 1). Dogs received one injection of meloxicam (0.1 mg/kg, subcutaneously) 24 h before surgery and another injection immediately after surgery. Each dog was kept in a single cage, wearing Elizabethan collars. The wound area was disinfected daily using iodophor. Hydrogen gas (99.99%, *v/v*) produced from a hydrogen gas generator (QL-500; Saikesaisi Hydrogen Energy Co., Ltd., Shandong, China) was bubbled into 400 mL of distilled water at the rate of 500 mL/min for 10 min to achieve saturation. All dogs had free access to standard food after postoperative fasting; the control group was provided DW (400 mL) thrice daily and the experimental group was provided saturated HRW (400 mL) thrice daily. Each dog had only one hour at a time to drink to keep the concentration of HRW constant, deprived of water in another time.

Skin wound samples were collected at 7, 14, 21, and 28 days after surgical operation, intubation, analgo-sedation, and monitoring; the surgical procedures were identical to the skin wound model, the skin was sutured with reduced-tension suture. The surgical site was daily disinfected by iodophor. The skin wound at A1 and B1 was sampled on day 7; A2 and B2 were sampled on day 14, A3 and B3 were sampled on day 21, A4 and B4 were sampled on day 28. All samples were picked with a scalpel. Following sampling, the wound was closed using a suture and sterilized with iodophor. A1, A2, A3, and A4 samples were fixed in 4% paraformaldehyde (20 g paraformaldehyde powder dissolved in 500 mL PBS buffer, followed by constant stirring using a magnetic stirrer. NaOH was used to adjust the solution pH). B1, B2, B3, and B4 samples were stored in a refrigerator at −80 °C.

### 2.3. Examination of the Skin Wound

The changes in the wounds in the HRW and the DW groups were observed on days 7, 14, 21, and 28 after the operation for bleeding. Observations included the presence or absence of bleeding, inflammatory exudate characteristics, wound healing, and the presence or absence of wound infection. The wounds were photographed on days 3, 7, 14, and 21, and wound was quantified using the ImageJ software (http://imagej.net/ImageJ, accessed on 15 August 2019).
Wound healing rate = (S0 − Sn) ÷ S0 × 100%

Sn: the wound area at 7, 14, 21, or 28 days, S0: the wound area at 0 day.

Dog skin wound healing was measured until the wound was completely covered by granulation tissue; the time of this process was the time of wound healing.

Paraffin sections were prepared for histopathological examination. H&E staining was used for evaluating epithelial thickness and neovascularization and Masson staining for collagen fiber. Briefly, five sections were chosen from each dog, and ten visual fields were randomly selected from different areas in each section. The evaluation of section was performed by two independent and blinded observers.

Oxidative stress was assessed by detecting malondialdehyde (MDA) and super oxide dismutase (SOD) levels in skin tissues following the instructions on the detection kits (Jiancheng Biotech Nanjing, Nanjing, China).

Total RNA was isolated from skin tissue using TRIzol Reagent (Invitrogen, Carlsbad, CA, USA). The RNA was reverse transcribed using Super Script First Strand cDNA System (Invitrogen, Carlsbad, CA, USA), and qRT-PCR was performed using the Light Cycler 480 Real-Time PCR System (Roche Diagnostics, Mannheim, Germany). Table 1 provides the list of primer sequences.

### 2.4. Statistical Analysis

All data were statistically analyzed using independent-sample *t*-test using SPSS 22.0 statistical software.

## 3. Results

### 3.1. Wound Healing

#### 3.1.1. Clinical Wound Evaluation

Figure 2 shows the presence of a significant difference in skin wound area between the HRW and DW groups between days 3 and 21. On day 3, the skin wound showed obvious hemorrhage, edema, and inflammatory reaction in the DW group and was not obvious in the HRW group. On day 7, there were abundant granulation tissue, newborn epithelia, and wound surfaces were humid and glossy in the HRW group, whereas the DW group had fewer granulation tissues and newborn epithelia, and wound surfaces were dried and matte, with large blood scab crust and with inflammatory exudate. On day 14, skin tissue defects were filled with granulation tissue in all dogs; the HRW group showed better re-epithelialization than the DW group with the appearance of scar tissue and hair growth. On day 21, wounds were almost fully re-epithelialized in the HRW group. On day 28, the wounds in the HRW group showed complete wound closure compared to incomplete closure in the DW group.

#### 3.1.2. Wound Healing Rate and Average Healing Time

During the experimental period, the rate of skin wound healing in the DW and HRW groups increased gradually until complete healing was achieved (Figure 3A, Table 2). On day 3, the HRW group (17.57 ± 0.55%) had a significantly higher rate of skin wound healing compared with that of the DW group (1.59 ± 0.55%) (*p* < 0.05). On day 7, the HRW group (46.82 ± 4.29%) had a significantly higher rate of skin wound healing compared with that of the DW group (16.44 ± 6.43%) (*p* < 0.01). The average duration of wound healing (Figure 3B, Table 3) was (19.00 ± 0.36 d) in the HRW group, which was significantly different than the DW group (22.00 ± 1.10 d) (*p* < 0.05).

### 3.2. Histopathology

#### 3.2.1. Wound Length

The results from the histopathological analysis of skin (Figure 4) show that on day 7, the wound length was still large and the surface of wound was covered with the granulation tissue in DW and HRW groups. On day 14, the skin wound length in the HRW group was significantly smaller and showed better re-epithelialization than the DW group (*p* < 0.05).

#### 3.2.2. Re-Epithelialization

The formation of neoepithelium was observed along the wound surface margin in the HRW group, initiating keratinocyte differentiation, epidermal cell differentiation, epidermis development, skin development, and keratinization. On day 14, undifferentiated keratinocytes were found in the DW group. On day 21, the HRW group showed marked reduction in epithelial flattening, gland formation, and epithelium thickness, whereas the DW groups showed irregularly thick epithelium (Figure 5A). Re-epithelialization were finished in the HRW group by day 28, with appearance of connective tissue, which indicated the appearance of scar tissue. On day 14, the average epidermal thickness was significantly lower in the HRW group (153.42 ± 31.72 μm) than the DW group (193.28 ± 45.80 μm) (*p* < 0.05; Figure 5B). On day 21, the average epidermal thickness was significantly lower in the HRW group (54.16 ± 2.53 μm) compared with the DW group (123.54 ± 46.14 μm) (*p* < 0.01). On day 28, all skin wounds in the canines had been healed and there was no significant difference between the HRW and DW groups regarding the average epidermal thickness.

#### 3.2.3. Neovascularization

Both HRW and DW groups showed the highest number of neovascular structures on day 7, which gradually decreased with time (Figure 5A,C, Table 4). On day 7, compared with the DW group (8.60 ± 2.97), there was a significantly higher number of neovascular structures in the HRW group (17.60 ± 1.14) (*p* < 0.01). On day 21, compared with the DW group (3.20 ± 0.81), there was a significant difference in the number of neovascular structures in the HRW group (1.20 ± 0.84) (*p* < 0.01; Figure 5C, Table 4).

#### 3.2.4. Collagen Fiber

The collagen fibers were stained blue, and the muscle fibers were stained red after Masson’s trichrome staining. The results of the pathological section showed that the collagen fibers in two groups increased with the passage of time (Figure 6). The collagen fibers in the HRW group were configured regularly at day 21 and were similar to healthy skin until day 28. The DW group showed staggered and irregular collagen fibers between days 7 and 21 and were fewer than those in the HRW group. On day 28, there was a dense deposition of irregular collagen fibers in the DW group.

### 3.3. Oxidative Stress Detection

During the experimental period, the level of MDA in skin tissues (Figure 7A, Table 5) first increased (from day 7 to 14) and then decreased (from day 14 to 28) in both DW and HRW groups. On day 7, compared with the levels of MDA in the skin tissue in the DW group (31.05 ± 4.51 U/mg), the same in the HRW group (17.68 ± 2.48 U/mg) was significantly lower (*p* < 0.01). On day 14, the MDA levels in the DW group (41.40 ± 2.74 U/mg) were significantly higher than that in the HRW group (17.47 ± 2.82 U/mg) (*p* < 0.01). Between days 21 and 28, the MDA levels in the skin tissues dropped sharply, and there was no significant difference between the two groups (*p* > 0.05).

During the experimental period, the SOD levels in the skin tissues (Figure 7B, Table 6) increased from day 7 to 28 in DW group, whereas they first decreased (from day 7 to 14) and then increased (from day 14 to 28) in the HRW group. On day 7, compared with the SOD levels in the skin tissue in the DW group (12.36 ± 0.96U/mg), the same in the HRW group (22.82 ± 1.57 U/mg) was significantly higher (*p* < 0.05). On day 21, the SOD levels in the DW group (23.36 ± 4.04 U/mg) were significantly lower than that in the HRW group (31.23 ± 6.13 U/mg) (*p* < 0.05). On day 28, the SOD levels in the DW group (24.39 ± 2.87 U/mg) were significantly lower than that in the HRW group (33.78 ± 7.11 U/mg) (*p* < 0.05).

The gene expression of Nrf-2 in HRW/DW ratio was significantly different between days 7 and 21, and was significantly different on day 14 (Figure 7C). The gene expression of HO-1 in HRW/DW ratio was significantly different between days 7 and 21 (Figure 7D). The gene expression of NQO-1 in HRW/DW ratio was significantly difference between days 7 and 21, and was significantly different on day 14 (Figure 7E).

### 3.4. Growth Factor Detection

The gene expression of VEGF in HRW/DW ratio was significantly different between days 7 and 14 (Figure 8A). The gene expression of PDGF in HRW/DW ratio was also significantly different between days 7 and 14 (Figure 8B).

## 4. Discussion

In this study, the positive effect of HRW treatment on skin wound healing in dogs was determined, which was primarily attributed to its antioxidant potential. The gold-standard assessment of skin wound healing involves the visual inspection of the wound for surface epithelization and reduction in the skin wound size with time [26]. Recent studies have shown that estrogen might be involved in the wound-healing process at the molecular level [27,28,29]. Thus, the dogs used in this study were all intact males to avoid differing levels of estrogen. The HRW treatment regimen exhibited substantial effects between days 7 and 28; the area of skin wounds was always less than that in the DW group. HRW was found to facilitate the rapid formation of granulation tissue to shorten the healing time than the DW group, which was probably due to the fact that HRW promoted the formation of blood vessels in the early stage of wound healing as well as relatively rapid re-epithelialization [30]. Here, HRW upregulated the expression of antioxidant genes and growth factor genes, which promoted wound healing. Additionally, the wound length in the HRW group was also significantly shorter than that in the DW group (Figure 4) between days 7 and 14.

Collagen fibers is the main extracellular matrix component, which acts as a structural scaffold in tissues, regulating cell proliferation and migration during skin wound healing [31]. Thus, the content of collagen fibers is an important parameter to examine skin wound healing [10]. The HRW group showed that the collagen fibers within the skin wound site were observed to be more regular and higher in content compared to the DW group (from days 7 and 21, Figure 6). Upregulated gene expression in PDGF stimulated collagen synthesis and the release of collagenase [32] (Figure 8B). Thus, elevated gene expression of PDGF resulted in more collagen fibers in the HRW group than the DW group. It also explained why collagen fibers increased in skin tissue after treatment with HRW in skin wound healing and was probably one of the major principles involved in HRW-mediated skin wound healing.

During the early stages of skin wound healing, an inflammatory response followed by re-epithelialization of the wound area occurs along with the establishment of granulation tissue accompanied by neovascularization [33]. Neovascularization is also critical for wound healing and tissue repair [34]. Neovascularization provides nutrients and oxygen to the wound, supports keratinocyte migration, transports mesenchymal stem cells to the skin, and then supports wound regeneration [35]. Angiogenesis is a process orchestrated by multiple angiogenic factors, among which VEGF is an essential growth factor regulating the critical steps of angiogenic processes [36]. At the same time, PDGF regulates angiogenesis by recruiting and primes the pericytes [37], promoting skin wound healing. In our experiment, the gene expression of VEGF (Figure 8A) and PDGF (Figure 8B) in both groups showed that the HRW group exhibited an enhanced ability for neovascularization at an early stage of skin wound healing (0–7 days) and complied with the results of the number of neovascular in the tissue section (Figure 5A,C, Table 4). Therefore, it was plausible that the effect of HRW on angiogenesis in skin wounds was regulated by increasing the expression of VEGF and PDGF, promoting the formation of granulation tissue. Studies have shown that VEGF increases cartilage formation during the early stages of endochondral bone formation, leading to a significant enhancement in bone formation and bone healing [38]. Fan et al. found that in endometrial repair, VEGF blockade dramatically inhibited re-epithelialization [39]. Elevated VEGF expression could drive re-epithelialization of the skin wound. Therefore, elevated gene expression of VEGF and angiogenesis were the positive signals during the skin wound healing process.

Nrf-2 is a critical regulator of antioxidant response [40], which can avoid the oxidative trauma as well as the development of oxidative stress. MDA is a metabolite of lipid peroxidation of unsaturated fatty acids by oxygen free radicals and indirectly reflects the severity of damage induced by free radicals [12,31]. The changes in MDA levels reflect the changes in oxygen-free radical content in the tissues. As a scavenger of free radicals, the SOD activity also reflects the number of free radicals [41]. The antioxidant roles of HRW were central to promoting wound healing. In the present study, we found that HRW activated the Nrf-2 antioxidant pathway and influenced SOD activity and MDA levels in skin tissue of dogs.

The NQO-1 and HO-1 genes are the downstream targets of Nrf-2 and these two genes are regulated by Nrf-2 expression, which is regarded as one of the most important intracellular antioxidant mechanisms [42]. Research has shown that hydrogen molecules can activate the expression of Nrf-2 in lung tissue, thereby promoting the expression of HO-1 and NQO-1 [24]. In our study, it was demonstrated that HRW treatment increased the expression of Nrf-2 (Figure 7C, from days 7 to 21) and then elevated the expression of its downstream phase II metabolic enzymes HO-1 (Figure 7D, from days 7 to 21) and NQO-1 (Figure 7E, from days 7 to 21). Nrf-2 can trigger transcription of SOD, reducing the MDA content in the skin tissue [43]. In our experiment, from days 7 to 14, the MDA content in the DW group was significantly higher than that in the HRW group (*p* < 0.01), and the SOD activity in the HRW group was significantly higher than that in the DW group (*p* < 0.05) on days 7, 21, and 28. As one of the reactive intermediates from oxidation, MDA is a marker of fatty acid oxidation; its elevated levels might lead to impaired and delayed skin wound healing in rats [44,45]. Li et al. found that wound lipid peroxidation and enhancement of SOD expression promoted the wound healing of chronic ulcers [46]. In the diabetic model, increased proinflammatory cytokines and oxidative stress in the wound microenvironment resulted in the delay of wound healing [47]. This suggested that the activity of SOD reflected the oxidation levels of skin tissue, and lower oxidation level could avoid oxidative damage, causing faster healing. Ohsawa et al. reported that 2% hydrogen gas inhaled by animals could scavenge hydroxyl radical (OH) and peroxynitrite anion (ONOO-), thereby improving oxidative stress-induced cell injury [48]. SOD can catalyze the dismutation of superoxide anions into oxygen and hydrogen peroxide, and this step also requires the participation of hydrogen. HRW provides more hydrogen for SOD to participate in the scavenging of oxygen radicals; therefore, HRW scavenges the hydroxyl radicals and superoxide radicals by enhancing the SOD vitality, thereby reducing MDA content, promoting skin wound healing.

## 5. Conclusions

The therapeutic effect of oral intake HRW in dogs can increase the number of blood vessels in the granulation tissue, accelerating wound epithelization, shortening wound healing time. We also found that HRW could enhance the antioxidant capacity of the dogs, activating the expression of Nrf-2 signaling pathway, elevating the expression of MDA, VEGF, and PDGF, while decreasing the expression of SOD. The dogs showed no discomfort in following the treatment strategy. These results show that HRW is a safe and effective veterinary clinical treatment for dog skin wound healing.

## Figures and Tables

**Figure 1 vetsci-08-00264-f001:**
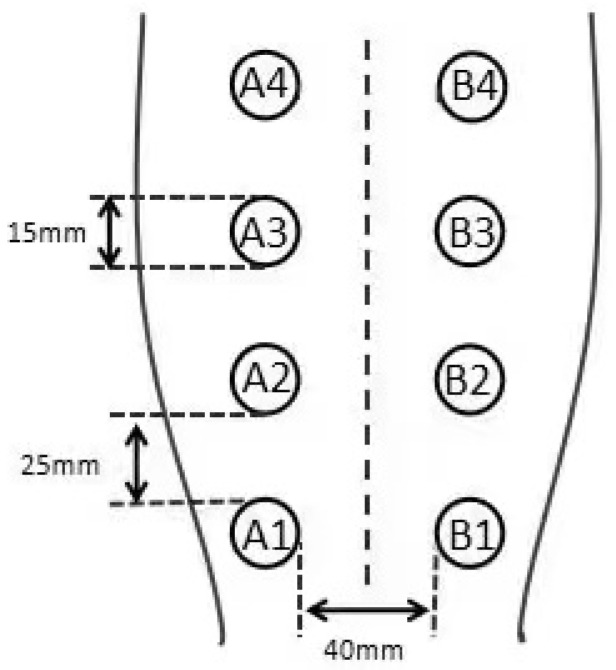
Four circular wounds with a diameter of 15 mm on each side of the midline of the shaved dorsum. Each wound is 20 mm from the midline of the dorsum, and the adjacent wounds are 25 mm from each other. From the tail side to the head side, and from left to right, the wounds are labeled in the following order: A1, A2, A3, A4, B1, B2, B3, B4.

**Figure 2 vetsci-08-00264-f002:**
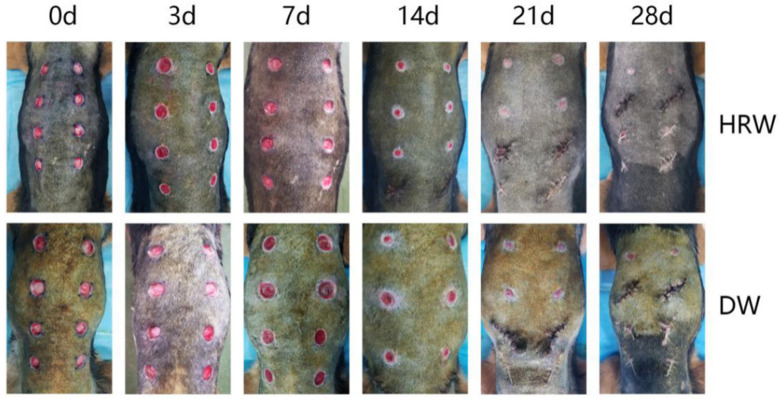
(HRW) Images of dogs in the experimental group at days 0, 3, 7, 14, 21, and 28 after modeling; (DW) Images of dogs in the control group at days(d) 0, 3, 7, 14, 21, and 28 after modeling. The same dog was photographed at each time point in HRW group both DW group.

**Figure 3 vetsci-08-00264-f003:**
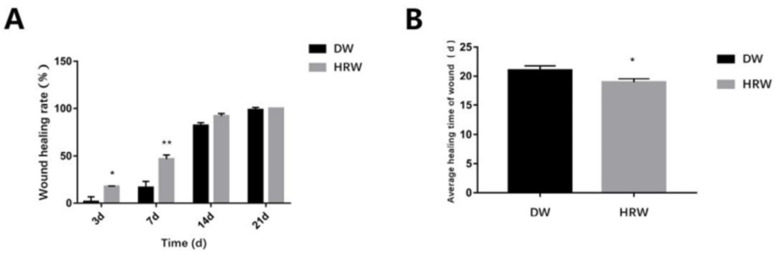
(**A**) The wound healing rate on days 3, 7, 14, and 21 between HRW and DW. (**B**)Skin wound was completely covered by the granulation tissue was considered the healing time of wound. Compared with two groups, * *p* < 0.05, ** *p* < 0.01.

**Figure 4 vetsci-08-00264-f004:**
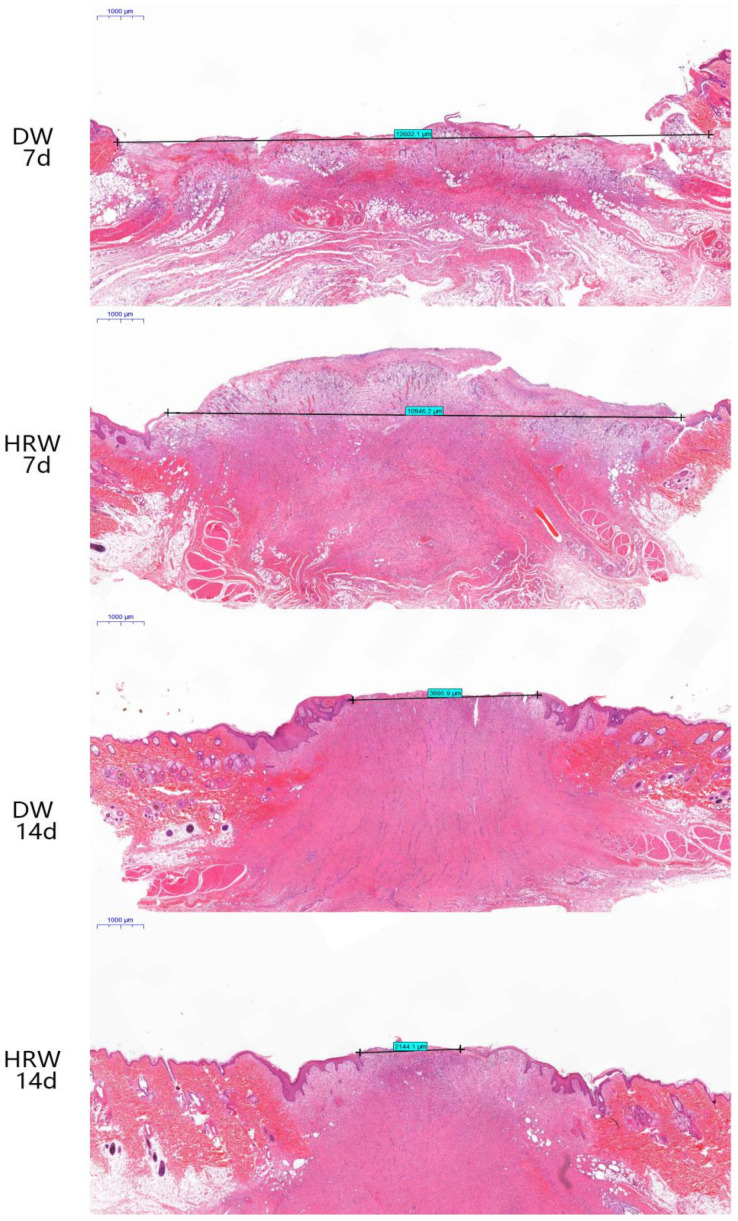
H&E staining was used to measure the length of wound of dogs in the HRW group on days 7 (10,946.2 μm) and 14 (2144.1 μm) and the DW group on days 7 (12,602.1 μm) and 14 (3895.9 μm).

**Figure 5 vetsci-08-00264-f005:**
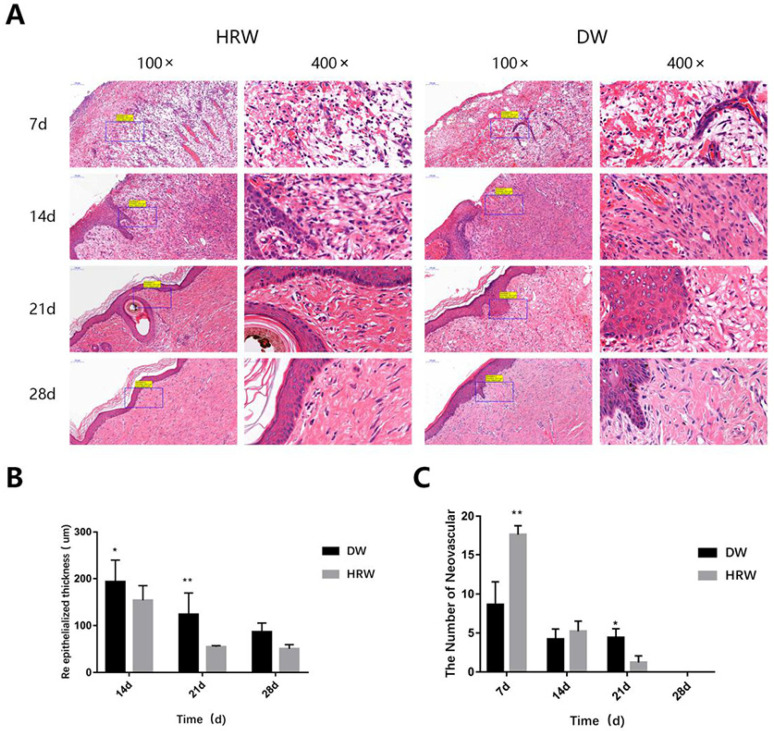
(**A**) H&E section of the skin wound. Angiogenesis, granulation tissue, and re-epithelialization were observed by H&E staining. (**B**) Re-epithelialized thickness. H&E samples were observed, and the epidermal thickness at the wound edge was measured using ImageJ. (**C**) Average number of blood vessels. The number of neovascular structures in skin tissue were assessed by H&E staining of the paraffin-embedded section. * *p* < 0.05, ** *p* < 0.01.

**Figure 6 vetsci-08-00264-f006:**
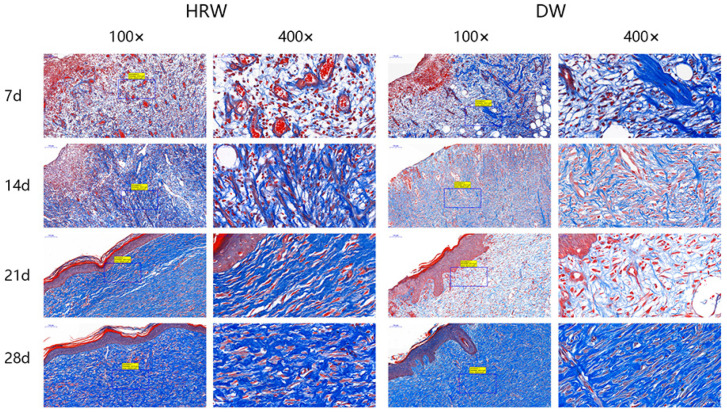
Skin wound paraffin sections were subjected to Masson staining. The collagen fibers were stained blue, and the muscle fibers were stained red after Masson’s trichrome staining.

**Figure 7 vetsci-08-00264-f007:**
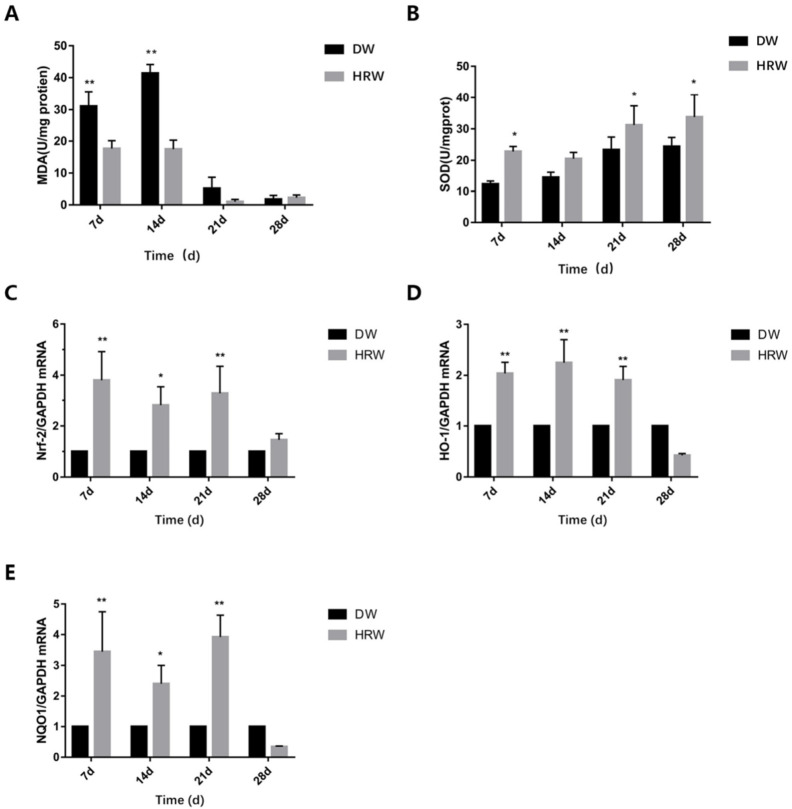
(**A**) MDA and (**B**) SOD activity of sample was determined using test kits. Relative (**C**) Nrf-2, (**D**) HO-1 and (**E**) NQO-1 mRNA levels were measured by TaqMan real-time PCR. * *p* < 0.05, ** *p* < 0.01.

**Figure 8 vetsci-08-00264-f008:**
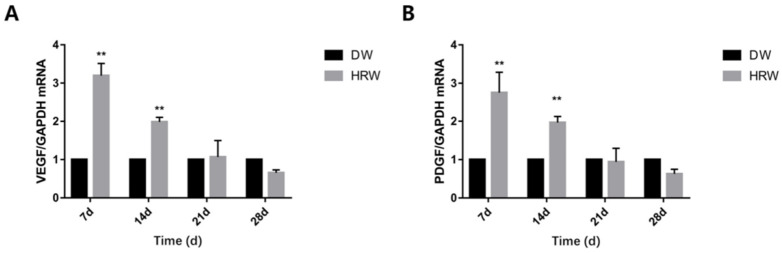
Relative (**A**) VEGF and (**B**) PDGF mRNA levels were measured by TaqMan real-time PCR. ** *p* < 0.01.

**Table 1 vetsci-08-00264-t001:** Primer sequence.

Gene	Gene Sequence Numbers	Primer Sequence (5’ to 3’)	Size (bp)
GAPDH	NM_008084.2	Forward: TCATGAGGCCCTCCACGAT Reverse: GATGGGCGTGAACCATGAG	157
NRF-2	NM_009925.4	Forward: CCCATCGGAAACCAGTGCAT Reverse: CATCTACGAACGGGAATGTCTCTG	150
HO-1	GI: 678541	Forward: GCCAAGACTGCCTTCCTGCTG Reverse: ACCCGTTGTCGTAGCCCTGAG	113
NQO1	NM_008607.2	Forward: ACCTGTACGCCATGAACTTCAACC Reverse: CTTCTGCTCGGCCACGATGTC	149
VEGF	NM_008607.2	Forward: GCCTTGCCTTGCTGCTCTACC Reverse: CTTCGTGGGGTTTGTGCTCTCC	84
PDGF	NM_008607.2	Forward: CCTGGCGTGCAAGTGTGAGAC Reverse: CCGAATGGTCACCCGAGTTTGG	112

**Table 2 vetsci-08-00264-t002:** The Wound healing rate (%) in DW and HRW groups at days 3, 7, 14, and 21. * *p* < 0.05, ** *p* < 0.01.

Time	DW	HRW
3 d	1.59 ± 5.04	17.57 ± 0.55 *
7 d	16.44 ± 6.43	46.82 ± 4.29 **
14 d	82.03 ± 3.09	92.07 ± 2.58
21 d	98.57 ± 2.45	100.00

**Table 3 vetsci-08-00264-t003:** The wound healing time (d) in DW and HRW groups. * *p* < 0.05.

Group	DW	HRW
Healing time	21.53 ± 1.68	19.25 ± 0.43 *

**Table 4 vetsci-08-00264-t004:** The number of neovascular structures in DW and HRW groups at days 7, 14, 21, and 28. * *p* < 0.05, ** *p* < 0.01.

Time	DW	HRW
7 d	8.60 ± 2.97	17.60 ± 1.14 **
14 d	5.20 ± 1.31	4.20 ± 1.41
21 d	3.20 ± 0.81 *	1.20 ± 0.84 *
28 d	0	0

**Table 5 vetsci-08-00264-t005:** The level of MDA (U/mg) in DW and HRW groups at days 7, 14, 21, and 28. ** *p* < 0.01.

Time	DW	HRW
7 d	31.05 ± 4.51 **	17.68 ± 2.48
14 d	41.40 ± 2.74 **	17.74 ± 2.82
21 d	5.20 ± 3.49	1.03 ± 0.71
28 d	1.78 ± 1.21	2.27 ± 0.84

**Table 6 vetsci-08-00264-t006:** The level of SOD (U/mg) in DW and HRW groups at days 7, 14, 21, and 28. ** *p* < 0.01.

Time	DW	HRW
7 d	31.05 ± 4.51 **	17.68 ± 2.48
14 d	41.40 ± 2.74 **	17.74 ± 2.82
21 d	5.20 ± 3.49	1.03 ± 0.71
28 d	1.78 ± 1.21	2.27 ± 0.84

## Data Availability

The data presented in this study are contained within the article.

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
