# Peer review of "The Therapeutic Effects of Oral Intake of Hydrogen Rich Water on Cutaneous Wound Healing in Dogs"

_vetsci, 2021, doi:10.3390/vetsci8110264_

Round 1

Reviewer 1 Report

The revised manuscript is OK for acceptance now

Author Response

Thank you very much for helping us to revise our manuscript entitled ‘The therapeutic effects of oral intake of hydrogen rich water on cutaneous wound healing in dogs’ in your busy schedule. Thank you very much for your great support for our research. Finally, thank you very much for your hard work.

Reviewer 2 Report

Reviewer comments for manuscript ID vetsci-1445161 ‘The therapeutic effects of oral intake of hydrogen rich water on cutaneous wound healing in dogs’

General comments

A nice effort by the authors for further improvement of the manuscript. It appropriately represents the excellent work done by the researchers. All my suggestions/corrections have been incorporated. There are very few errors which I have pointed out specifically. I recommend the publication of the manuscript.

Specific comments

Line 26: Please rewrite ‘accelerate wound epithelization, reduce inflammatory reaction, stimulate’ as ‘accelerated wound epithelization, reduced inflammatory reaction, stimulated’

Line 137: Please rewrite ‘The items to be observed’ as ‘ Observations included’

Author Response

Thank you very much for helping us to revise our manuscript entitled ‘The therapeutic effects of oral intake of hydrogen rich water on cutaneous wound healing in dogs’ in your busy schedule. We have revised the manuscript according to your suggestions (Line: 26-27, 139). Thank you for your comments concerning our manuscript.

Reviewer 3 Report

No further comments.

Author Response

Thank you very much for helping us to revise our manuscript entitled ‘The therapeutic effects of oral intake of hydrogen rich water on cutaneous wound healing in dogs’ in your busy schedule. Thank you very much for you appreciation. Finally, thank you very much for your hard work.

Reviewer 4 Report

The manuscript has significantly improved since its first version. I only suggest to better detail the Captions of tables, to enable all readers to maximize the information presented.  

Author Response

Thank you very much for helping us to revise our manuscript entitled ‘The therapeutic effects of oral intake of hydrogen rich water on cutaneous wound healing in dogs’ in your busy schedule. According to your suggestion, we have redrafted  the captions of tables (Line: 192, 200, 240, 277, 294). Very grateful to the reviewers' comments, which is helpful for improving our manuscript.

This manuscript is a resubmission of an earlier submission. The following is a list of the peer review reports and author responses from that submission.

Round 1

Reviewer 1 Report

This research topic is interesting and the authors have tried to provide some results about this topic. Reasonable revisions are needed before being considered for acceptance, Comments and suggestions: 1, the structure of this manuscript need to be reorganized, the logistics need to be improved 2, are there any published refs about hydrogen water for wound healing? 3, more background information about wound healing are suggested to be provided, these refs about be helpful, such as: J. Mater. Chem. B, 2021,9, 6738-6750; Biomater. Sci., 2021,9, 4388-4409. 4, how to make sure the different body sites receive the same amount of hydrogen? 5, how to make sure the hydrogen water is properly and effectively (keep the hydrogen content) taken by the dog? 6, the writing quality need to be improved

Reviewer 2 Report

The authors present a very interesting paper in the field of wound healing in dogs, using the innovative approach of hydrogen-rich water. Reference list is up-to-dated and contains all relevant papers. Despite this, I have some concerning about the structure and robustness of the manuscript. In materials and methods some points (detailed below) need to be clarified. Result section is not easy to follow, and extensive rephrasing and tables addition is needed to clarify the manuscript. Discussion section is too long and difficult to follow.

In my opinion, the manuscript cannot be accepted in the present form and requires major revisions.

Page 1, abstract: please rewrite the result section (lines 19-23) to be more clear in these statements.

Page 1, lines 31-32. This sentence contains a repetition, please rephrase.

Page 2, lines 58-60: authors introduce four methods to produce HRW but only three are exposed.

Page 4, examination of the wound: it would have been worth to apply also a scoring system for wound healing (i.e. Modified Draize and/or Hollander scores) to obtain more data to be statistically analyzed. In addition, it would have been interesting to use a 3D measurement, imaging and documentation system that provides precise measurement and healing trends.

Page 4, lines 138-139: do you mean when granulation tissue reached the same level of normal skin around open wounds?

It would have been interesting to assess also the healing rate of these wounds in terms of depth (i.e. how much granulation tissue was present at each time point? How was the depth of wounds at each timepoints?)

Page 5, figure 2. What is the orientation of the pictures? I may hypothesize that they are upside-down since the only open wounds are those on the upper part of each pictures, and following what authors stated in Materials and -methods section, the last sampled wounds should be those close to the tail

Page 6, 3.1.2. Wound Healing Rate and Average Healing Time. Please insert these data in a table to allow an easy reading and comparison. Same for 3.2.3. Neovascularization and 3.3. Oxidative Stress Detection on page 8.

No mention regarding the HRW intake in experimental dog. It would have been interesting to have data about the average individual water intake for each dog, for both HRW and DW.

Reviewer 3 Report

Reviewer comments for manuscript ID vetsci-1370569 entitled ‘The effects of drinking hydrogen-rich water on skin wound healing in dogs’

General comments

It is a novel study on the therapeutic effects of oral administration of hydrogen rich water on skin wound healing. The authors have done a commendable effort in the design, implementation, and execution of the study. 

Reviewer comments for manuscript ID vetsci-1370569 entitled ‘The effects of drinking hydrogen-rich water on skin wound healing in dogs’

General comments

It is a novel study on the therapeutic effects of oral administration of hydrogen rich water on skin wound healing. The authors have done a commendable effort in the design, implementation, and execution of the study. The presentation is also excellent despite some errors in the grammatical aspects of the writing.

In the introduction, the authors must explicitly introduce this therapy in veterinary practice clearly stating the gaps in the literature. Material methods are nicely presented. Results are well written and substantiated with excellent photographs, figures, and graphs. I am very impressed with the thorough and precise discussion. However, the conclusions drawn are very vague and do not do justice to the hard work done in the results and discussion. It must conclude with the future projections of this work and limitations, if any.  

Specific comments

I suggest a different title of the manuscript ‘The therapeutic effects of oral intake of hydrogen rich water on cutaneous wound healing in dogs’

Lines 12-17: Please do not place sub-headings in the abstract e.g., background, results, discussion.

Line 20: Please replace ‘Histopathological results’ with ‘Histopathology’

Line 21: Please insert ‘in the ‘between that and HRW group.

Line 31-33: Please reframe this sentence for clarity.

Lines 33-34: Please reframe this sentence as ‘Skin wounds caused by diverse elements are routinely presented in a clinical veterinary setting’

Lines 34-37: I am sorry, I am not able to understand this sentence. Please clarify.

Lines 84-85: I think we can write this ‘Breeding Base of Harbin’ as ‘a breeder based in Harbin’

Lines 91-92: Please delete these lines as they are already covered under animal ethics permission that have been previously stated.

Line 94: Please replace ‘No animals’ with ‘None of the animals’

Line 96: Please replace ‘Surgical Operation’ with ‘Creation of surgical wounds’

Lines 108-09: Won’t this affect the results of your experiment? Please clarify.

Figure 1: Nice representation!

Lines 121-23: Please reframe this sentence. It’s not clear.

Line 135: Please provide details of the software in brackets.

Lines 159-61: Please delete.

Lines 267-69,273-74: Please delete ‘extremely’

Lines 292-93: You must mention this in the materials and methods section.

Lines 296-306: Please justify your assertion by providing appropriate references.

Lines 325-26: Please complete the sentence.

In the introduction, the authors must explicitly introduce this therapy in veterinary practice clearly stating the gaps in the literature. Material methods are nicely presented. Results are well written and substantiated with excellent photographs, figures, and graphs. I am very impressed with the thorough and precise discussion. However, the conclusions drawn are very vague and do not do justice to the hard work done in the results and discussion. It must conclude with the future projections of this work and limitations, if any.  

Specific comments

I suggest a different title of the manuscript ‘The therapeutic effects of oral intake of hydrogen rich water on cutaneous wound healing in dogs’

Lines 12-17: Please do not place sub-headings in the abstract e.g., background, results, discussion.

Line 20: Please replace ‘Histopathological results’ with ‘Histopathology’

Line 21: Please insert ‘in the ‘between that and HRW group.

Line 31-33: Please reframe this sentence for clarity.

Lines 33-34: Please reframe this sentence as ‘Skin wounds caused by diverse elements are routinely presented in a clinical veterinary setting’

Lines 34-37: I am sorry, I am not able to understand this sentence. Please clarify.

Lines 84-85: I think we can write this ‘Breeding Base of Harbin’ as ‘a breeder based in Harbin’

Lines 91-92: Please delete these lines as they are already covered under animal ethics permission that have been previously stated.

Line 94: Please replace ‘No animals’ with ‘None of the animals’

Line 96: Please replace ‘Surgical Operation’ with ‘Creation of surgical wounds’

Lines 108-09: Won’t this affect the results of your experiment? Please clarify.

Figure 1: Nice representation!

Lines 121-23: Please reframe this sentence. It’s not clear.

Line 135: Please provide details of the software in brackets.

Lines 159-61: Please delete.

Lines 267-69,273-74: Please delete ‘extremely’

Lines 292-93: You must mention this in the materials and methods section.

Lines 296-306: Please justify your assertion by providing appropriate references.

Lines 325-26: Please complete the sentence.

Reviewer 4 Report

Dear authors

See comments on attach pdf.

Overall, the experimental setting and results are interesting and well stablished.

However there is need for english editing as many phrases are not clear or repeated.

There is confusion too on the results, as some of them do not correspond to what it is shown on the figure.
